# Exploring Plastome Diversity and Molecular Evolution Within Genus *Tortula* (Family Pottiaceae, Bryophyta)

**DOI:** 10.3390/plants14172808

**Published:** 2025-09-08

**Authors:** Hamideh Hassannezhad, Mahmoud Magdy, Olaf Werner, Rosa M. Ros

**Affiliations:** 1Department of Plant Biology, Faculty of Biology, University of Murcia, Campus of Espinardo, 30100 Murcia, Spain; hamideh.h@um.es (H.H.); werner@um.es (O.W.); 2Department of Genetics, Faculty of Agriculture, Ain Shams University, Cairo 11241, Egypt

**Keywords:** chloroplast genomics, phylogenetic relationships, SSR, molecular markers, genetic diversity, mosses

## Abstract

The Pottiaceae family represents one of the most diverse and ecologically adaptable bryophytes; however, its chloroplast genome diversity remains largely unexplored. This study aimed to investigate plastome variation and identify evolutionary informative loci within the moss genus *Tortula*. We performed a comprehensive comparative plastome analysis of nine species within the genus *Tortula*, using *Syntrichia princeps* as an outgroup within the family Pottiaceae. High-quality chloroplast genomes were assembled and annotated based on next-generation sequencing (NGS) data. All plastomes exhibited conserved quadripartite structures with genome size ranging from 121,889 to 122,697 bp. Adenine–thymine (AT)-rich dinucleotide repeats were the most abundant simple sequence repeats (SSRs), and several genes contained unique higher-order SSRs, suggesting potential utility as population-level markers. Codon usage analysis revealed species-specific biases, particularly in leucine, serine, and threonine codons, with *Tortula acaulon* exhibiting the most pronounced deviation. Phyloplastomic analysis based on maximum likelihood identified two major clades, indicating that *Tortula* section *Tortula* is not monophyletic. Several highly informative loci were found to replicate the full plastome phylogenetic signal. Additionally, a subset of genes, including *atp*E and *mat*K, exhibited nonsynonymous-to-synonymous substitution (dN/dS) ratios that suggest possible positive selection. These findings provide new insights into chloroplast genome evolution within *Tortula*, while identifying candidate loci for future phylogenetic and evolutionary studies. This study contributes to a more robust understanding of plastome-based studies in Pottiaceae and highlights efficient molecular markers for broader bryophyte phylogenomics.

## 1. Introduction

Bryophytes—including mosses, liverworts, and hornworts—are key terrestrial plants that enhance soil moisture, contribute to nutrient cycling, and support soil formation [1]. Mosses, the most diverse group, are characterized by leafy gametophores, mostly spirally arranged leaves with a central midrib, and distinctive sporophytes with sporangia that usually open via an apical lid, revealing hygroscopic peristome teeth that aid spore dispersal [2], and they thrive in diverse environments [3]. Within mosses, the Pottiaceae family (over 1500 species) exhibits exceptional ecological adaptability, with traits favoring survival in arid and semi-arid habitats [4,5]. As one of the most desiccation-tolerant bryophyte lineages, Pottiaceae often dominate dry-ground vegetation globally [6].

*Tortula* Hedw. is a genus of small acrocarpous mosses within the family Pottiaceae. Its name derives from the Latin *tortus* (“twisted”) and the diminutive suffix -*ula*, referring to the twisted peristome teeth [7]. However, not all species currently placed in *Tortula* share this feature. Zander [5] significantly revised the genus by downplaying the taxonomic weight of sporophytic traits such as capsule type and peristome structure. He proposed these features to reflect a morphological reduction series, rather than serving as reliable characters for generic delimitation. Based on this perspective, he expanded *Tortula* to include taxa previously treated as distinct genera, including *Desmatodon* Brid., *Phascum* Hedw., portions of *Pottia* Ehrh. ex Fürnr., and *Protobryum* (Dicks.) J. Guerra & M.J. Cano. Zander [5,8] also proposed the segregation of several morphologically distinct lineages into new or reinstated genera, such as *Chenia* R.H. Zander, *Dolotortula* R.H. Zander, *Hennediella* Paris, *Hilpertia* R.H. Zander, *Sagenotortula* R.H. Zander, *Stonea* R.H. Zander, and *Syntrichia* Brid. Although initially controversial, Zander’s taxonomic framework has since received empirical support from molecular phylogenetic analyses, which corroborate the monophyly of the revised genus [9,10,11,12,13]. The molecular phylogeny using chloroplast *rps4* gene sequences has further refined relationships within *Tortula* and related genera [12]. This study confirms that *Syntrichia* is a distinct monophyletic clade, whereas *Pottia* appears polyphyletic, with some of their species better placed in *Tortula*. Similarly, some species of *Desmatodon*, *Phascum cuspidatum* Hedw. and *Protobryum bryoides* (Dicks.) J. Guerra & M.J. Cano should be included in *Tortula*. Conversely, species previously assigned to *Tortula*, such as *T. rhizophylla* (Sakurai) Z. Iwats. & K. Saito, should be transferred to the genus *Chenia*. The close relationship of *Crossidium*, *Pterygoneurum*, and *Stegonia* with *Tortula* were also confirmed.

The number of species currently assigned to the genus *Tortula* is estimated to range between 144 and 165 [4,5,14,15]. Zander [5] classified these species into four infrageneric groups at the sectional level: section *Tortula*, which includes the type species and most of the traditionally recognized members of the genus; section *Pottia* (Ehrh. ex Reichenb.) Kindb., for which the present valid name is section *Cuneifoliae* (Schimp.) Ochyra [16], comprising mainly species formerly assigned to *Desmatodon*, *Phascum*, *Pottia*, and *Protobryum*; section *Schizophascum* (Müll. Hal.) R.H. Zander, which includes additional species previously placed in *Phascum* and *Pottia*; and section *Hyophilopsis* (Cardot & Dixon) R.H. Zander, also comprising some species formerly included in *Pottia*.

*Tortula* species are found in a wide range of habitats, mainly on soil [5], but also on rocks, acidic tree bases, and artificial substrates such as walls; they occur across a broad elevational range, from lowland regions to alpine zones [17]. *Tortula* is ecologically significant, as it represents the most species-rich genus of Pottiaceae in many regions, particularly in arid and semi-arid environments such as the Mediterranean Basin, where up to 38 species have been recorded [18]. Some individual countries within this region, such as Italy and Spain, host between 28 and 30 species [19,20], a number comparable to the total *Tortula* diversity reported for all North America, which includes 29 species [15].

Chloroplasts are essential organelles in plants that convert solar energy into chemical energy through photosynthesis, resulting in the production of adenosine triphosphate (ATP) and carbohydrates, which are vital for plant growth and metabolism [21,22]. The chloroplast genome, or plastome, encodes numerous essential proteins that are crucial for photosynthesis and other metabolic pathways [23,24,25]. Typically, plastomes are circular DNA molecules that carry the genetic information necessary for chloroplast function and provide valuable insights into the evolutionary history of the host organism [26]. Due to their conserved structure, maternal inheritance, and relatively low mutation rates compared to nuclear genomes, plastomes have become fundamental tools in plant systematics, evolutionary biology, and phylogenetics [27]. The plastome typically exhibits a quadripartite structure consisting of two identical inverted repeat (IR) regions that separate a large single-copy (LSC) region and a small single-copy (SSC) region [25,28]. The IR regions often contain ribosomal RNA (rRNA) genes, while the LSC and SSC regions house most coding DNA sequence (CDS) and transfer RNA (tRNA) genes involved in photosynthesis and other chloroplast functions [29]. While the basic structure of the plastome is conserved across plant species, there are notable variations in genome size and nucleotide sequences, even among species within the same genus [27,30].

A significant challenge in the molecular genetic study of mosses is their small size and limited biomass, which often restricts the quality and quantity of DNA obtainable for next-generation sequencing (NGS). This issue can be mitigated through *in vitro* cultivation of moss spores, which are harvested from mature capsules collected in the wild [31]. Recent advances in NGS, particularly Illumina sequencing technology, have greatly enhanced the ability to perform high-throughput assembly and comparative analysis of plastomes. These improvements allow for the detection of structural variation, gene rearrangements, and evolutionary divergence across plant lineages [32]. Despite such advancements, plastome diversity within the Pottiaceae family remains underexplored. The limited number of complete chloroplast genomes available in public databases [33,34,35,36,37] constrains the development of efficient molecular markers for phylogenetic and taxonomic applications.

This study presents a comprehensive comparative evolutionary analysis of chloroplast genomes from nine species of the genus *Tortula*, representing the two most important and species-rich sections of the genus with broad geographical distribution. The studied species were *Tortula acaulon* (With.) R.H. Zander, *T. atrovirens* (Sm.) Lindb., *T. brevissima* Schiffn., *T. lindbergii* Broth., *T. mucronifolia* Schwägr., *T. muralis* Hedw. (var. *aestiva* Brid. ex Hedw.), *T. pallida* Lindb., *T. protobryoides* R.H. Zander, and *T. subulata* Hedw. For comparative purposes, *Syntrichia princeps* (De Not.) Mitt., a species belonging to a sister genus, was included as an outgroup to provide a wider view of chloroplast genome diversity outside the *Tortula* genus within the Pottiaceae family. Utilizing high-throughput Illumina sequencing technology, high-quality chloroplast genomes were assembled to evaluate interspecific variation and evolutionary divergence within the genus *Tortula*. The primary aim of this study is to assess plastome diversity in *Tortula* and identify highly informative genomic regions that replicate full plastome phylogenetic signals. These loci may serve as cost-effective molecular markers for future taxonomic studies, without the technical demands of complete NGS-based plastome sequencing.

## 2. Results

### 2.1. Organelle Genome Characteristics

Among *Tortula* species, the total plastome length of the moss species examined in this study varied between 121,889 bp in *T. muralis* and 122,697 bp in *T. brevissima* (med = 122,523 bp and mean = 122,442 ± 233 bp). All plastomes displayed the typical quadripartite circular structure characteristic of land plant chloroplast genomes. The LSC region ranged from 83,376 bp in *T. muralis* to 84,173 bp in *T. brevissima* (med = 83,974 bp and mean = 83,912 ± 226 bp). The SSC region showed slight variation, ranging from 18,422 bp in *T. acaulon* to 18,684 bp in *T. brevissima* (med = 18,606 bp and mean = 18,598 ± 74 bp). The IR regions ranged from 9920 bp in *T. brevissima* to 10,006 bp in *T. protobryoides* (med = 9975 bp and mean = 9966 ± 32 bp). Modest differences were observed between the LSC, SSC, and IR regions, possibly indicating region-specific functional or evolutionary influences. All plastomes were assembled into a single circular contig, with high coverage (>404× on average). Annotation via Chlorobox identified a total of 126 genes, of which 81 were protein-coding genes, 37 tRNAs, and 8 rRNAs, per plastome, with no variation in gene number across species. Gene order and synteny were conserved, and no structural rearrangements were detected. The hypothetical gene *ycf*2 was the only gene that showed major variation in the frame length across species (mean = 6002 ± 228 bp; Appendix A). The guanine–cytosine (GC) content across the species examined ranged from 28.20% in *T. subulata* to 28.50% in *T. atrovirens* and *T. pallida*, with slight variation detected among the plastome regions (±0.1%) (Figure 1, Table 1 and Appendix A).

### 2.2. Tandem Repeats—Microsatellites

Based on a comparative scheme across the species, the polymorphic microsatellite profiling across the studied nine species from the genus *Tortula* and the outgroup *S. princeps* revealed both conserved patterns and species-specific variation in simple sequence repeats (SSRs) composition (Figure 2). Dinucleotide repeats were by far the most abundant across all species, with the AT motif dominating consistently. The number of AT repeats ranged from 190 in *T. pallida* to 204 in *T. subulata*, highlighting their genomic abundance and evolutionary conservation. The AG motif was the second most frequent dinucleotide, showing minimal variation among species (29–33 repeats), whereas CG repeats were rare and relatively uniform (2–3 repeats per species), consistent with their generally low frequency in eukaryotic genomes. Trinucleotide SSRs, although less abundant than dinucleotide repeats, revealed a slightly more informative pattern. The AAT motif was the most frequent trinucleotide across all species, with counts ranging from 72 in *T. mucronifolia* to 84 in *T. acaulon*. Among other repeats, *T. atrovirens* and *T. mucronifolia* displayed the highest amounts of AAG (eight copies), while *T. brevissima* exhibited the highest AGC count (six), and *T. muralis* showed the highest ACT motif count (three).

The three species with the highest number of AAAT repeats, listed from lowest to highest, were *T. brevissima* (14), *T. subulata* (16), and *T. mucronifolia* (17). Tetranucleotide repeats revealed clearer interspecific variation, with the AAAT motif being the most dominant. Notably, *T. lindbergii* displayed unique SSR motifs such as AAGG and higher counts of AAAG and AAAC compared to other species, distinguishing it from the rest. In addition, the AACT motif was exclusively found in *T. pallida*, further highlighting species-specific occurrences of certain tetranucleotides. Pentanucleotide and hexanucleotide repeats were less frequent but showed important taxonomic signatures. *Tortula brevissima* stood out with the highest number of pentanucleotide motifs (seven in total), including four copies of AAAAG. The AATAT motif, though rare, was only detected in *T. brevissima*, *T. muralis* and *S. princeps*. For hexanucleotide repeats, *T. lindbergii* was the only species with the AAAAAG motif, while *T. brevissima* exhibited the broadest diversity of hexameric repeats, including AAAAAT, AAAATT, and AAATTT. Higher-order repeats were extremely rare but diagnostic. *Tortula lindbergii* uniquely harbored the nine-nucleotide repeat AAAAGAACT, while *S. princeps* was the only species to contain the AAATATAAT motif. These long, complex SSRs may serve as species-specific molecular markers due to their uniqueness and low likelihood of homoplasy.

### 2.3. IR Boundaries Contraction and Expansion

The IR regions undergo contraction and expansion, a common phenomenon contributing to genome size variation among species. This dynamic process is reflected in structural differences observed at the junctions between IRs and single-copy regions (Figure 3). At the JLB junction (LSC/IRb), the distance of *rps*7 from the boundary varied slightly; for example, it was 1057 bp in *T. protobryoides*, 1107 bp in *T. subulata*, and 1046 bp in *S. princeps*.

The JSB junction (IRb/SSC) showed slight differences in the positioning of the *chlL* gene, which was present in all species. While *chl*L was generally located in the SSC region, it extended partially into the IRb region. The extent of this overlap varied across species, ranging from 97 bp in *T. acaulon* to 180 bp in *T. brevissima*.

The JSA junction (SSC/IRa) involved the *ndh*F gene, which extended into the IRa region to varying extents. The overlap ranged from 4 bp in *T. acaulon* to 71 bp in *T. brevissima* and 3 bp in *S. princeps* as an outgroup. The JLA junction (IRa/LSC) was characterized by the presence of *trn*M near the junction. The *trn*M gene was consistently located at the IRa/LSC boundary in all species, ranging from 60 bp in *T. atrovirense* to 94 in *T. acaulon*, while *rpl*23 remained in the LSC region. As a result, the IR boundary structures among *Tortula* species exhibit a combination of overall conservation and lineage-specific variation, particularly at the JLB, JSB, JSA, and JLA junctions.

### 2.4. Relative Synonymous Codon Usage (RSCU) and Codon Bias

To investigate synonymous codon usage bias and its interspecific variation, RSCU values were computed for 59 codons encoding 20 amino acids and three stop signals across ten species of the family Pottiaceae (Figure 4, Appendix A). Codon usage bias, reflected by deviation from the neutral RSCU value of 1.0, was evident for nearly all amino acids, with several codons showing strong preference or avoidance across various species groups. In general, the RSCU profiles reveal a consistent trend of strong codon usage bias across species, with *T. acaulon*, *T. brevissima*, and *T. mucronifolia* exhibiting the most deviated patterns among all the species studied, especially in high RSCU values for specific codons.

In the amino acids with six synonymous codons, leucine (L) exhibited the strongest codon usage bias among all amino acids. The TTA codon was highly overrepresented in *T. pallida* (3.445), *T. lindbergii* (RSCU = 3.447), and *T. protobryoides* (3.478), with *T. acaulon* showing the highest RSCU value in the dataset (4.079). In contrast, CTG and CTC were severely underrepresented, with RSCU values as low as 0.05 (*T. acaulon*) and 0.098 (*T. muralis*), indicating extreme codon preference divergence. Arginine (R) demonstrated a moderate but consistent codon usage bias. AGA was the most frequently used codon in all species, with *T. muralis* exhibiting the highest RSCU (2.751). The CGG codon was universally the least favored, particularly in *T. acaulon* (RSCU = 0.192). Serine (S) showed substantial interspecies variation in codon usage. TCT was the most favored serine codon, especially in *T. acaulon* (RSCU = 2.369) and *T. atrovirens* (1.858). AGC, while not strongly preferred, was relatively more common in *T. subulata*, *T. protobryoides*, and *T. brevissima* (RSCU = 0.970–0.974), but was notably underused in *T. acaulon* (RSCU = 0.256), highlighting considerable variation across taxa.

Among amino acids with four synonymous codons, alanine (A) exhibited the least variability in codon usage. GCT was consistently the preferred codon in all species (RSCU > 1.9), peaking in *T. acaulon* (2.244). Threonine (T) demonstrated moderate variability, as ACT and ACA were generally preferred, with *T. acaulon* showing a particularly high preference for ACT (RSCU = 2.073). ACG was underrepresented across species, with *T. acaulon* again displaying the lowest usage (RSCU = 0.141). Valine (V) showed the most notable variation within this group. GTT was strongly preferred in all species (RSCU > 2.0), in contrast to *T. atrovirens* where GTA was in balance with GTT, while GTC and GTG were consistently underrepresented.

The three stop codons usage patterns were generally consistent but revealed species-specific peaks. TAA was dominant in all species, with a particularly high RSCU in *T. acaulon* (2.537). TGA was consistently the least used stop codon (RSCU = 0.199 across species). TAG usage showed moderate variability, ranging from 1.105 in *T. atrovirens* to 0.265 in *T. acaulon*.

Amino acids with two synonymous codons, such as glutamic acid (E), exhibited a clear codon preference in *T. acaulon*, where GAA was strongly favored (RSCU = 1.859) and GAG was significantly underused (RSCU = 0.141). Phenylalanine (F) showed the most distinct bias among two-codon amino acids. TTT was consistently favored, especially in *T. acaulon* (RSCU = 1.838), while TTC was substantially underused (RSCU = 0.162). Lysine (K) and asparagine (N) showed comparable species-specific biases. AAA and AAT were consistently preferred across taxa, while AAG and AAC were underrepresented. In *T. acaulon*, this bias was especially pronounced (AAC = 0.239; AAG = 0.106), reflecting strong selection for specific codons in this species. Tyrosine (Y) followed a similar trend, with TAT preferred across all species and TAC underrepresented, particularly in *T. acaulon* (RSCU = 0.233).

### 2.5. Identification of Hypervariable Regions and Barcode Candidates

#### 2.5.1. Haplotype and Nucleotide Diversity in the Chloroplast Genome

The full plastome alignment from nine *Tortula* species and *Syntrichia princeps* were dissected to 241 comparable regions, including 123 genetic regions (CDS, rRNA and tRNA), and 118 intergenic spacers (IGS). Regions with less than 10 bp, were discarded from the analysis. DNA polymorphism was assessed for each region (Table 2 and Appendix A). The mean region length was 459.41 ± 633.80 bp, with 31.50 ± 45.90 segregating sites (S) and a mean number of haplotypes (Hap) of 7.17 ± 3.24. The average haplotype diversity (Hd) across all regions was 0.79 ± 0.32, and the overall nucleotide diversity (π) was ~0.03 ± 0.02, reflecting a moderate level of polymorphism across the plastome.

The CDS regions exhibited the highest number of aligned sites, averaging 837.92 ± 820.39 bp. These regions harbored a moderate level of polymorphism, with the number of S averaging 51.72 ± 63.57, and Hap reaching 8.12 ± 2.50. Hd was high (0.90 ± 0.18), while π was relatively low (~0.02 ± 0.01), indicating shared haplotypes with small sequence differences (Table 2).

Compared to CDS, the IGS regions displayed moderate sequence length (263.41 ± 313.47 bp) and a polymorphism profile like that of CDS. The average number of S was 27.64 ± 33.58, with 7.86 ± 2.78 Hap. Hd remained high (0.87 ± 0.25), and π was slightly higher than that of CDS (~0.04 ± 0.02), suggesting that IGS regions are more variable at the nucleotide level despite their shorter length.

Compared to other regions, the rRNA genes were the longest, with an average aligned length of 1130.75 ± 1197.19 bp, but they were the least polymorphic. These genes had a low number of S (8.00 ± 11.29) and Hap (4.50 ± 3.42), with correspondingly low Hd (0.5 ± 0.42) and negligible π (~0.00 ± 0.00). These findings reflect the conserved nature of rRNA genes.

The tRNA regions were the shortest among all types (~164.38 ± 229.11 bp) and exhibited limited polymorphism, with 5.79 ± 11.59 S and 3.18 ± 2.99 Hap. The Hd was modest (0.35 ± 0.36), and π remained low (~0.01 ± 0.02), further indicating the conserved functionality of tRNA genes.

The selection criteria were based on identifying regions with the maximum Hap (up to 10) while maintaining genetic neutrality, as indicated by non-significant Tajima’s D values. This approach yielded a total of 90 candidate regions, comprising 49 IGS, 38 CDS, and three tRNA genes (Appendix A).

#### 2.5.2. Phylogenetic Analyses

The phylogram consisted of trees derived from complete plastomes demonstrating strong concordance across various genomic datasets. To evaluate phylogenetic signal strength, all hypervariable loci (Hap = 10) were tested individually in addition to the entire structural regions comprising the LSC, SSC, and IRs (Figure 5 and Appendix A). These individual loci included *atp*E, *atp*I, *cem*A, *chl*N, *psb*B, *mat*K, *ndh*A, *ndh*D, *ndh*D-*trn*L, *rbc*L, *rpo*B, *rpo*C1, and *rps*7-*trn*V.

The Maximum Likelihood (ML) and Bayesian Inference (BI) analyses using *S. princeps* as the outgroup separated the nine *Tortula* species with strong bootstrap support values of 1.0 at all nodes. Two major subclades were defined: subclade I included group A that consisted of *T. brevissima* and *T. muralis*. Subclade II started with *T. atrovirens* followed by group B, which comprised *T. subulata* and *T. mucronifolia*, followed by group C that included *T. pallida* and *T. lindbergii*. In parallel, *T. protobryoides* and *T. acaulon* formed another closely related pair (group D). Group A, *T. atrovirens*, and group B collectively represent members of the *Tortula* section *Tortula*. Together, group C and group D represent a distinct evolutionary lineage corresponding to section *Cuneifoliae*. Consequently, section *Cuneifoliae* appears to be monophyletic, whereas section *Tortula* is shown to be non-monophyletic. These groupings underscore the phylogenetic coherence within the genus. The loci *atp*E, *cem*A, *chl*N, *psb*B, *mat*K, *ndh*A, *ndh*D-*trn*L, *rbc*L, *rpo*B, *rpo*C1, and *rps*7-*trn*V showed strong phylogenetic signals and high reliability in reconstructing evolutionary relationships consistent with the full plastome-based phylogenetic tree. While the chloroplast genome was generally consistent across the taxa, two loci, specifically *atp*I and *ndh*D (highlighted in red in Figure 5), exhibited differences in the position of *T. atrovirens* (positioned between cluster B and C, or clustering with clade B, respectively). These were not 100% consistent with the full plastome tree but still fully clustered the other species successfully in accordance with the full plastome phylogenetic signal.

#### 2.5.3. Nonsynonymous-to-Synonymous Substitutions Ratio (dN/dS Ratio) or ω

To explore the selective pressures acting on chloroplast protein-coding genes, the dN/dS ratio or ω across aligned coding sequences was calculated. In contrast, a smaller subset of genes showed ω > 1, suggesting signals of positive selection. Among these, *ycf*2, *ndh*G, and *yc*f1 exhibited the highest ω values, strongly indicating adaptive evolution. Other genes, including *cem*A and *ycf*66, also showed elevated ω values, pointing to potential functional shifts driven by environmental pressures. A group of genes including *mat*K, *rpl*32, *ndh*K, *rpo*C2, *atp*E, *rps*3 and *Paf*I also fell within the range of positive selection but were situated near the neutrality threshold (ω ≈ 1), suggesting possible relaxed functional constraints or episodic positive selection.

Importantly, *atp*E and *mat*K, which were also identified as informative markers in our phylogenetic analysis, showed ω values close to neutrality. This reinforces their potential relevance in both phylogenetic resolution and adaptive divergence. Most genes exhibited ω < 1, which indicates that purifying selection is the predominant evolutionary force preserving chloroplast gene function. These results highlight candidate genes for further evolutionary and functional investigation (Figure 6).

## 3. Discussion

Comparative analysis of plastid genomes revealed a high degree of structural conservation among the nine *Tortula* species studied, with *S. princeps* serving as an outgroup for comparative context. Only minor differences in genome size and GC content were observed, while the canonical quadripartite structure, gene order, and gene content were consistently maintained. These patterns highlight strong evolutionary constraints on chloroplast genome architecture and suggest that plastome structure and function remain highly stable across closely related moss taxa, as well as other plant species [24].

The total plastome lengths, ranging from 121,889 bp to 122,697 bp, fall within the typical range for bryophyte plastid genomes (120,000–130,000 bp), consistent with previous findings by Chang and Graham and Goffinet et al. [38,39]. The observed size differences were primarily attributed to variation in the LSC region, and IR regions, with only minor differences noted in the SSC region. Among the species examined, the LSC was the most variable region. Given that it encodes the majority of plastid genes, this variation likely results from subtle expansions in intergenic spaces rather than from large-scale structural rearrangements [40].

SSR profiling across *Tortula* genus plastomes revealed the dominance of dinucleotide repeats, particularly the AT motif, consistent with the AT-rich composition of bryophyte plastid genomes [41]. The frequent presence of AT repeats and the rarity of CG motifs further reflect this compositional bias, which is typical of plastid DNA [21]. While dinucleotide motifs were broadly conserved across species, notable species-specific variation was observed in trinucleotide and tetranucleotide repeats, particularly in *T. lindbergii* and *T. pallida*. This variation highlights their potential as informative molecular markers for population analysis studies [42]. Their application in conservation genetics is particularly valuable in mosses, where intraspecific variation plays a crucial role in guiding preservation strategies [23,28,41]. However, further validation is necessary, as only one accession per species was analyzed in this study—due to technical constraints in obtaining sufficient DNA for NGS from moss axenic gametophytes. This limitation is less critical when applying SSR-based techniques at the population scale, which typically require lower DNA input. Expanding SSR-based studies to include more taxa and ecological contexts will enhance their application in bryophyte systematics and support more targeted, evidence-based conservation and adaptation efforts [43,44].

The expansion and contraction of IR boundaries are recognized as key factors in chloroplast genome size variation [45]. In the genus *Tortula*, IR regions displayed high size conservation (9920 bp to 10,006 bp), consistent with the structural stability reported in other land plants [30,46]. Similarly, GC content remained stable across species and lower than that of most vascular plants, aligning with patterns observed in moss plastomes [11,27,34]. This low GC bias may reflect evolutionary constraints linked to genome compactness and metabolic adaptation in bryophytes [47]. Although overall plastome architecture was conserved, subtle shifts at IR boundaries suggest lineage-specific variation, potentially serving as indicators of evolutionary divergence [48].

Our analysis of RSCU across nine *Tortula* species and *S. princeps* revealed a pronounced bias toward A/T ending codons, especially TTA (leucine), TCT (serine), and ACT (threonine), with *T. acaulon* exhibiting the most extreme codon preferences. This bias is consistent with the AT-rich composition typical of moss plastomes. Although codon usage patterns have not been specifically characterized for *Syntrichia ruralis* (Hedw.) Gaertn., Meyer & Scherb. (formerly known as *Tortula ruralis* (Hedw.) G. Gaertn., B. Mey. & Scherb.), its plastome structure and AT-rich composition are consistent with the codon usage trends observed in our study, supporting the likelihood of conserved A/T ending-biased codon preference within this lineage [30]. Furthermore, structural variation and genome diversity in these plastomes may be influenced by recombination-dependent replication and gene conversion mechanisms, as described by Ruhlman et al. [46]. Notably, the exceptionally high frequencies of TTA and TAA codons in *T. acaulon* indicate lineage-specific divergence within the genus, which may reflect adaptation to unique ecological conditions or ongoing evolutionary differentiation. The underrepresentation of GC-rich codons such as CTG, CTC, and ACG further reflects mutational bias influencing codon usage [45,49]. Such codon preferences might contribute to translational efficiency in resource-limited environments typical of bryophytes and may reflect adaptive divergence among species with differing ecological niches.

Compared with other bryophytes, including species from the genus *Marchantia* L., *Physcomitrium patens* (Hedw.) Bruch & Schimp., and *Rhodobryum giganteum* (Schwägr.) Paris, codon bias patterns suggest that selective pressures on translational efficiency and genome composition are evolutionarily active among bryophytes [24,50,51]. These data highlight the utility of RSCU profiling in revealing both conserved evolutionary trends and species-level variation, underscoring its value for genomic studies in *Tortula* and related bryophytes [45].

Our results showed moderate nucleotide diversity and high haplotype diversity across the chloroplast genome, consistent with observations in other Pottiaceae species such as *S. ruralis*, where non-coding regions exhibit elevated variation [30,34]. IGS regions displayed higher nucleotide variability than coding regions, reflecting patterns reported in *P. patens* [51], while rRNA and tRNA genes remained highly conserved, as seen in other bryophytes [50]. We identified several hypervariable loci (i.e., *atp*E, *cem*A, *chl*N, *mat*K, *ndh*A, *ndh*D*-trn*L, *psb*B, *rbc*L, *rpo*B, *rpo*C1, and *rps*7*-trn*V) that correspond exactly with plastome-wide evolutionary relationships. These loci improve upon the limited resolution of commonly used chloroplast markers such as *rps*4, *trn*L, and *trn*H*–psb*A [32,43,48,52]. Although nuclear markers like nrITS have been employed to investigate gene flow and polyploidy, and *rps*4 is commonly used in phylogenetic studies [12,13,53], these universal markers often struggle to distinguish closely related species within Pottiaceae [13]. Our plastome-wide approach, supported by recent comprehensive studies, highlights the value of lineage-specific markers with higher resolution, which the hypervariable regions identified here provide, offering enhanced tools for species identification and evolutionary analyses.

Our results indicate several loci that have been shown to be effective in broader bryophyte phylogenetic studies (e.g., *rbc*L and *mat*K). Importantly, the genes *atp*E and *mat*K displayed close-to-neutral ω (dN/dS ratio) compared to others, suggesting their potential relevance in both phylogenetic reconstruction and studies of adaptive molecular evolution. Our phylogenetic analysis does not reflect the segregation of two of the sections proposed by Zander [5], namely, section *Tortula* and section *Cuneifoliae*. Only section *Cuneifoliae* appears to be monophyletic, suggesting that the current sectional classification within *Tortula* may not accurately reflect evolutionary relationships. Although the number of *Tortula* species included in this study is limited compared to the total number of species in each section, the lack of monophyly in section *Tortula* implies that morphological criteria alone are insufficient for reliable classification. A re-evaluation incorporating plastome-based markers could lead to revised sectional delimitation, in light of additional species with a correct loci selection.

Previous phylogenetic studies, such as Werner et al. [12,13], utilizing chloroplast *rps*4 sequences, also highlighted the paraphyly and polyphyly within the genus, particularly noting that species historically grouped under *Tortula* often do not form a single cohesive clade. Similar to our findings, Werner et al. [12] observed conflicting placements of certain taxa based on different loci, emphasizing the locus-specific nature of phylogenetic signals, even though, in our study, *rps*4 is not the most suitable phylogenetic marker to study this genus. Most loci, such as *atp*E, *mat*K, and *rbc*L, but not *rps*4 [12], are congruent with the full plastome tree, supporting their utility in resolving relationships. However, *T. atrovirens* showed inconsistent placement in trees depending on the locus, with some genes (e.g., *atp*I and *ndh*D), positioning it differently, which may be influenced by locus-specific evolutionary effects. In contrast to our reported chloroplast version of *T. atrovirens*, a previously reported poorly assembled plastome of the species documented gene missingness for 12 genes, including *pet*N, that are involved in photosynthetic electron transport [45], potentially confounding phylogenetic inference for this species and underscoring the importance of high-quality plastome assemblies. To overcome such discrepancies, it is important to analyze multiple accessions, species or conduct pan-plastome comparisons [23] before drawing conclusions about significant functional divergence or developing efficient molecular markers. Future studies should expand taxon sampling and include multiple accessions *per* species to better account for intraspecific variation and confirm marker robustness across populations.

This study successfully addressed its original aim of identifying highly variable plastome regions in *Tortula* species. Through full plastome comparison and phylogenetic signal testing, we validated several loci—including *atp*E and *mat*K—as promising molecular markers, improving the resolution beyond traditional barcoding genes. These findings support a more robust framework for taxonomic and evolutionary studies in Pottiaceae.

## 4. Materials and Methods

### 4.1. Sampling and Axenic In Vitro Cultivation

Specimens from the Pottiaceae family were collected across various habitats in southern and central Spain. The sampling focused on multiple species of the genus *Tortula*, as well as some species of the genus *Syntrichia* for use as outgroup species in phylogenetic analyses. Plants bearing mature and closed capsules were selected for axenic *in vitro* cultivation, following the protocol of Magdy et al. [42], with the aim of producing sufficient sterile plant material for sequencing. One to two accessions *per* species were established in culture to ensure adequate material for downstream genomic work. The final list of species that showed enough biomass for DNA extraction at adequate concentration for NGS sequencing are listed in Appendix A. Photographs of the species can be found in in the Bryologia gallica web page (http://bryologia.gallica.free.fr/, accessed on 25 August 2025) and the British Bryological Society web page (https://www.britishbryologicalsociety.org.uk, accessed on 25 August 2025).

### 4.2. DNA Extraction and Sequencing

Genomic DNA was extracted using the cetyltrimethylammonium bromide (CTAB) method following Murray and Thompson [54]. The integrity and concentration of the extracted DNA were assessed using spectrophotometry and agarose gel electrophoresis. This procedure was further optimized based on the protocol established by Werner et al. [55] for bryophytes, ensuring high-quality DNA suitable for downstream molecular applications. High-quality chloroplast genomes were obtained using Illumina sequencing technology. Paired-end sequencing (2 × 150 bp) was performed at ~30× coverage on the Illumina NovaSeq 6000 platform by Novogene (Munich, Germany).

### 4.3. Plastome Assembly and Annotation

The raw sequencing reads were initially assessed for quality using FastQC [56] to evaluate overall quality metrics, including per-base quality scores, GC content, and adapter content. When the sample passed the quality check, adapter trimming and removal of low-quality bases (Phred score < 20) were performed with Trimmomatic [57], ensuring that only high-quality reads were retained for downstream assembly.

*De novo* assembly and genome annotation were conducted following the single-contig approach described by Magdy et al. [27], with some modifications. Clean, high-quality reads were assembled using the single-contig approach in Geneious Prime 2023.1.1, resulting in a complete, circular plastome contig for each sample. To confirm and refine assembly accuracy, a second *de novo* assembly was performed in which the initial contigs were used as trusted contigs in SPAdes v3.15.5 [58] (implemented within Geneious), applying multiple k-mer values (21, 45, 65, 85, 105). Circularity and full-length genome structure were confirmed visually in Geneious.

To ensure sequence accuracy, trimmed reads were iteratively re-mapped to the assembled plastome in five consecutive rounds, correcting potential base-calling errors through consensus generation. Assembly completeness was confirmed by verifying the presence of all expected chloroplast regions (LSC, SSC, and IRs), with IR boundaries identified using Repeat Finder in Geneious.

Plastome annotation was performed using GeSeq [59], referencing *Syntrichia ruralis*, and manually curated in Geneious to ensure accurate gene boundary definitions. Annotated maps were visualized using OGDRAW [60].

### 4.4. Repetitive Sequences and Tandem Repeats Analysis

SSRs, including both perfect and compound microsatellites, were detected using the Phobos plugin in Geneious Prime (Biomatters Ltd., Auckland, New Zealand), enabling accurate screening across complete chloroplast genomes [61]. SSR motifs ranging from mono- to hexanucleotide repeats (1–10 bp) were identified using the following minimum repeat thresholds: 10 for mononucleotides, 6 for dinucleotides, and 5 for tri-, tetra-, penta-, and hexanucleotides. This is consistent with standard criteria for chloroplast SSR analysis. The resulting SSR datasets were processed in Microsoft Excel. Visualization of motif abundance and distribution were used to generate stacked area plots that illustrate the relative frequencies of different SSR motif types across species.

### 4.5. IR Boundaries Analysis

Comparative analysis of the IR boundaries among *Tortula* species and the outgroup *S. princeps* was conducted using Geneious software. The boundaries between the LSC, SSC, and IR regions were specifically examined at four junctions: JLB (LSC/IRb), JSB (IRb/SSC), JSA (SSC/IRa), and JLA (IRa/LSC). For precise visualization and comparison, IRscope [62] was used. While IR regions are typically among the most conserved components of chloroplast genomes, variations in overall plastome length often result from minor expansions or contractions at these junction sites.

### 4.6. Codon Usage Preference

Protein-coding genes were extracted using Geneious Prime. Codon usage bias was assessed using MEGA v11, where RSCU values were calculated to evaluate codon preference [63]. Codons with RSCU values greater than 1.0 were considered overrepresented, whereas those with values significantly less than 1.0 were considered underrepresented. Codon usage tables were generated to examine usage patterns across genes. Multi-panel dot plots were constructed by grouping codons according to their corresponding amino acids and plotting RSCU values across species in R environment using the ggplot2 package [64]. Codons were color-coded for clarity, and faceting was applied to display individual amino acid groups in separate panels.

### 4.7. Identifying and Evaluating Hypervariable Regions

#### 4.7.1. Haplotype and Nucleotide Diversity in the Chloroplast Genome

Geneious Prime was used to perform alignments and generate visualizations that facilitated the identification of candidate molecular markers within these divergent regions. To further explore structural variation, the mauve alignment tool [65] was employed for comparative genomic analysis. The alignment was dissected into regions for the CDS, IGS, tRNAs and rRNAs, including all ten species. For each region, S, Hap, Hd, and π were calculated using DnaSP v6.12.03 [66]. Hd was defined as the probability that two randomly selected haplotypes differ, while π represented the average number of nucleotide differences per site between two sequences [67]. Mean values and standard deviations for these parameters were computed for major genomic categories (CDS, IGS, rRNA, and tRNA) to assess the distribution of sequence polymorphism across the chloroplast genome [23].

Hypervariable regions were selected based on Hap = 10, reflecting a complete divergence level within a region, as each species possessed a distinct copy. These hypervariable regions were subsequently considered as potential targets for phylogenetic and evolutionary analyses.

#### 4.7.2. Phylogenetic Analyses

Complete chloroplast genome sequences, including the IR, SSC, and LSC regions, and hypervariable regions were realigned with the mauve genome aligner and further refined using Geneious Prime. Phylogenetic reconstruction was initially performed using FastTree v2 with the Generalized Time Reversible (GTR) model and default bootstrap settings to assess branch support (shown in decimal 0–1) [68]. *Syntrichia princeps* was included as an outgroup. Based on the same alignment, Bayesian phylogenetic inference was conducted using MrBayes v3.2.7a [69], based on the same aligned sequence dataset used for the Maximum Likelihood (ML) analysis. The GTR+G model of nucleotide substitution was applied. Two independent Markov Chain Monte Carlo (MCMC) runs were performed for 2,000,000 generations, sampling every 1000 generations, with the first 20% discarded as burn-in. Convergence was assessed using average standard deviation of split frequencies (<0.01). The resulting consensus tree with posterior probabilities was visualized using Geneious and compared with the ML tree.

#### 4.7.3. Nonsynonymous-to-Synonymous Substitutions Ratio (dN/dS Ratio) or ω

The dN/dS ratio, or ω, was used to evaluate selection pressure on chloroplast protein-coding genes. A value of ω > 1 indicates positive selection, ω = 1 suggests neutral evolution, and a ω < 1 reflects purifying selection. Protein-coding sequences were aligned using MAFFT v7.3 [70]. To ensure accurate codon-based comparison, stop codons and gaps were removed prior to alignment. The cleaned alignments were analyzed in MEGA 11, applying codon-based evolutionary models to estimate pairwise dN and dS values [63]. The resulting dN/dS ratios were visualized to highlight variation in selective pressure across genes, with positively selected genes identified by their elevated ω values.

## 5. Conclusions

This study provides important insights into the chloroplast genome architecture of *Tortula* (Pottiaceae), highlighting a combination of conserved structural features and lineage-specific divergence. While the plastome structure remains stable across species, variations in the LSC, SSC, and IR boundaries reflect evolutionary shifts that may relate to species adaptation and divergence. Among the hypervariable regions identified, *mat*K and *atp*E emerged as particularly informative markers, not only for resolving species-level relationships but also for their elevated dN/dS (ω) values, suggesting a potential role in adaptive molecular evolution. These two genes are strong candidates for future plastome-based phylogenetic and evolutionary studies. From technical point of view, additional loci of ~1.5 kbp such as *cem*A, *psb*B, *ndh*A, *rbc*L, and *chl*N also showed promise, providing a broader toolkit for molecular systematics and species identification as additional DNA barcodes. Additionally, microsatellite profiling revealed high interspecific variability, especially in tri- and tetranucleotide repeats, offering further resources for conservation genetics and population-level assessments in mosses. This work contributes novel comparative genomic data for an underexplored bryophyte group and provides clearly validated molecular tools to enhance species delimitation, phylogeny, and future conservation research. These findings lay a robust foundation for future taxonomic refinement in *Tortula* and the broader Pottiaceae family, especially when integrated with nuclear and mitochondrial datasets in further analyses.

## Figures and Tables

**Figure 1 plants-14-02808-f001:**
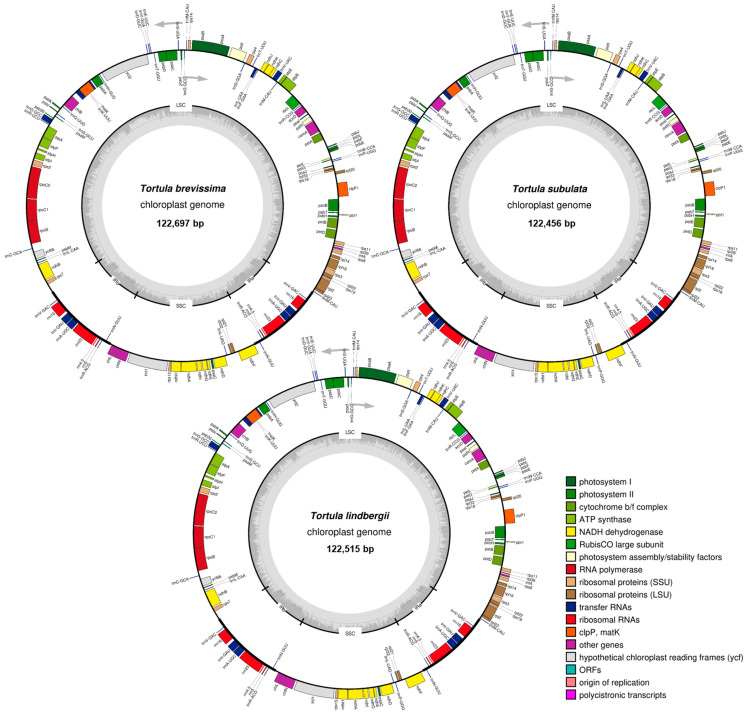
The complete chloroplast genome maps of three exemplary *Tortula* species include *T. brevissima* with a total length of 122,697 bp, *T. lindbergii* with 122,515 bp, and *T. subulata* with 122,456 bp. The inner circle represents the guanine–cytosine (GC) content and is divided into the major chloroplast regions: large single-copy (LSC), small single-copy (SSC), and inverted repeats (IRa and IRb). The outer circle represents the complete chloroplast sequence, with genes annotated on the outside of the circle indicating forward-strand genes and those on the inside indicating reverse-strand genes. Direction of transcription is indicated by the two grey arrows.

**Figure 2 plants-14-02808-f002:**
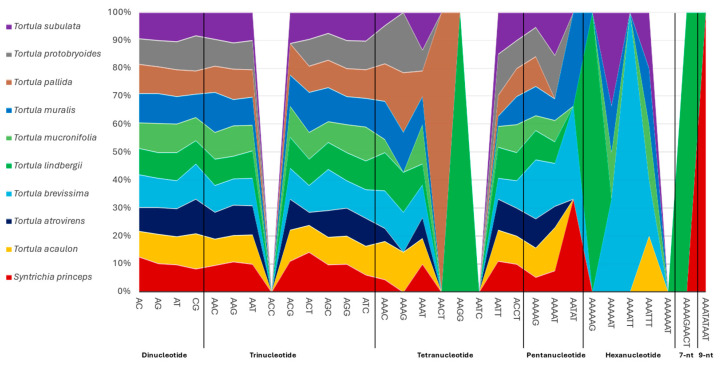
Proportional stacked area plot of simple sequence repeats (SSRs) motif frequencies across ten Pottiaceae species (nine *Tortula* species and *Syntrichia princeps*). Each motif’s relative abundance is represented along the *x*-axis, while the *y*-axis shows its cumulative proportion across species. Color bands represent individual species.

**Figure 3 plants-14-02808-f003:**
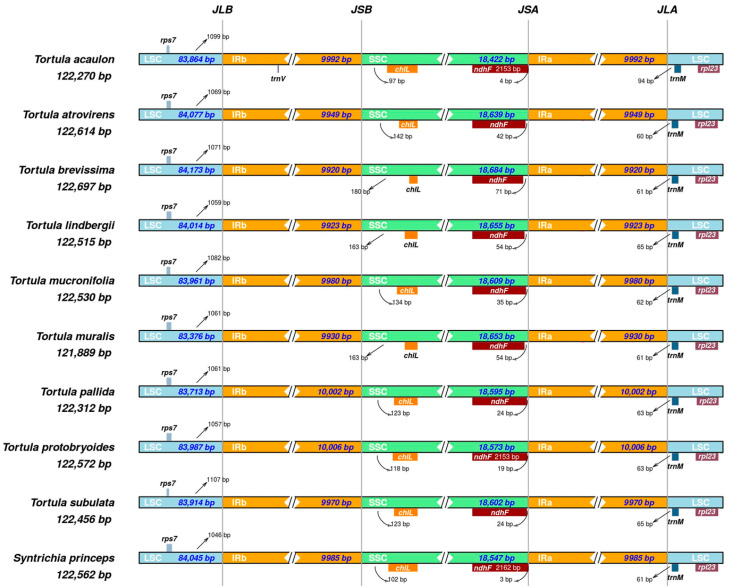
Comparisons of the large single-copy (LSC), inverted repeat a (IRa), small single-copy (SSC), and inverted repeat b (IRb) boundaries among ten Pottiaceae chloroplast genomes (nine *Tortula* species and *Syntrichia princeps* as an outgroup).

**Figure 4 plants-14-02808-f004:**
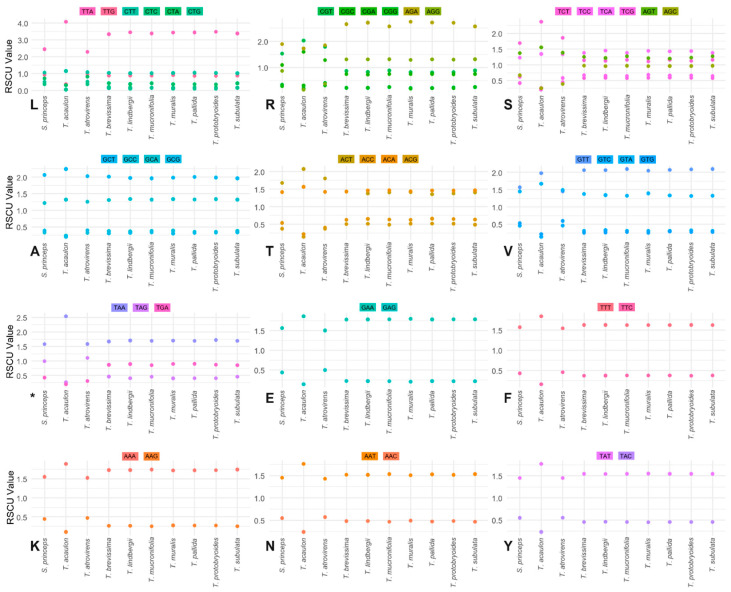
Relative Synonymous Codon Usage (RSCU) patterns across nine species of the genus *Tortula* and *Syntrichia princeps*. Dot plots show RSCU values for each codon grouped by amino acid [leucine (L), arginine (R), serine (S), alanine (A), threonine (T), valine (V), stop codon (*), glutamic acid (E), phenylalanine (F), lysine (K), asparagine (N), tyrosine (Y)]. Each point represents the RSCU value of a specific codon in each species, with codons distinguished by color. Species are presented along the *x*-axis. RSCU values greater than 1 indicate codons used more frequently than expected under equal usage, while values less than 1 indicate underrepresentation. The plots highlight both conserved and species-specific codon usage biases, particularly for amino acids with high degeneracies such as L, S and R.

**Figure 5 plants-14-02808-f005:**
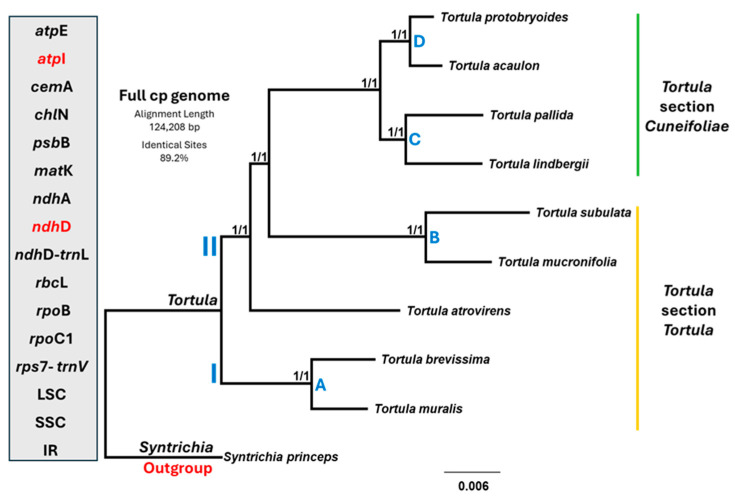
Maximum Likelihood (ML) phylogenetic tree constructed for nine *Tortula* species based on their chloroplast genomes, with *Syntrichia princeps* used as the outgroup. Node support is indicated by ML bootstrap followed by Bayesian posterior probabilities (ML/BI) (shown in decimal 0–1). Two well-supported major clades (bootstrap = 1/1) were identified. Clade I includes group A, comprising *T. brevisima* and *T. muralis*. Clade II contains *T. atrovirens* and the remaining species and is further subdivided into three groups (B–D) Groups A and B include species of *Tortula* section *Tortula* and groups C and D species of *Tortula* section *Cuneifoliae*. A box on the left lists the 16 chloroplast loci used independently in the analysis, including protein-coding genes (*atp*E, *cem*A, *chl*N, *psb*B, *mat*K, *ndh*A, *rbc*L, *rpo*B, *rpo*C1), intergenic regions (*ndh*D*-trn*L, *rps*7-*trn*V), and structural regions (LSC, SSC, IR). In addition, genes marked in red (*atp*I and *ndh*D) show variation associated with the phylogenetic placement of *T. atrovirens* (positioned between cluster B and C, or clustering with clade B, respectively). Scale indicates genetic distance.

**Figure 6 plants-14-02808-f006:**
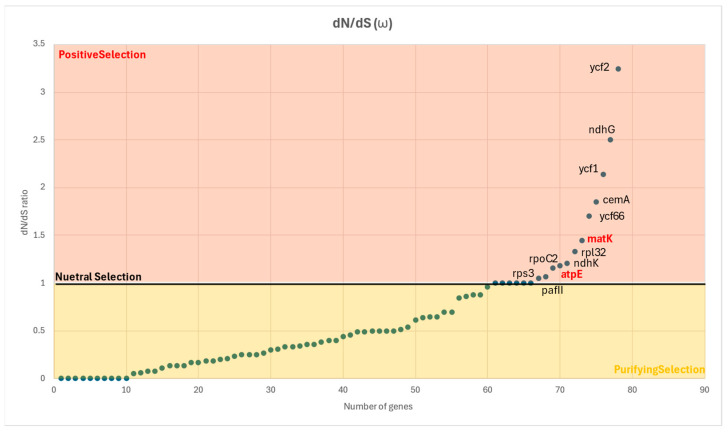
Distribution of nonsynonymous-to-synonymous substitutions ratio (dN/dS ratio) or ω across chloroplast protein-coding genes. Each dot represents a single gene, plotted by its dN/dS ratio. Most genes fall below the neutrality threshold (ω < 1), indicating strong purifying selection (yellow region). A smaller number of genes exhibit ω > 1, suggesting potential positive selection (orange region). Notable genes under positive selection include *ycf*2, *ndh*G, *ycf*1, *cem*A, and *ycf*66. Genes such as *mat*K, *rpl*32, *ndh*K, *rpo*C2, *atp*E, *rps*3, and *Paf*I cluster around the neutral selection boundary (ω ≈ 1), indicating possible relaxed constraints or episodic selection. Genes labeled in red (*atp*E and *mat*K) are of particular interest due to their dual roles in selection and phylogenetic resolution.

**Table 1 plants-14-02808-t001:** Basic characteristics of *Syntrichia princeps* and *Tortula* species chloroplast genomes (LSC, large single-copy region; SSC, small single-copy region; IR, inverted repeat regions; GC%, guanine–cytosine percentage; length of the regions expressed in base pairs, bp).

Species	LSC (bp)	SSC (bp)	IR (bp)	Total (bp)	GC%
*Syntrichia princeps*	84,045	18,547	9985	122,562	28.30%
*Tortula acaulon*	83,864	18,422	9992	122,270	28.40%
*Tortula atrovirens*	84,077	18,639	9949	122,614	28.50%
*Tortula brevissima*	84,173	18,684	9920	122,697	28.30%
*Tortula lindbergii*	84,014	18,655	9923	122,515	28.40%
*Tortula mucronifolia*	83,961	18,609	9980	122,530	28.20%
*Tortula muralis*	83,376	18,653	9930	121,889	28.40%
*Tortula pallida*	83,713	18,595	10,002	122,312	28.50%
*Tortula protobryoides*	83,987	18,573	10,006	122,572	28.40%
*Tortula subulata*	83,914	18,602	9970	122,456	28.20%

**Table 2 plants-14-02808-t002:** Summary of DNA polymorphism statistics across different chloroplast genome regions in nine species of the genus *Tortula* and *Syntrichia princeps*. Four genomic regions were analyzed: chloroplast coding sequences (CDS), intergenic spacers (IGS), ribosomal RNA (rRNA), and transferring RNA (tRNA). For each one, the mean number and their standard deviations (SD) of aligned sites (length in bp), segregating sites (S), haplotypes number (Hap), haplotype diversity (Hd), and nucleotide diversity (π) are given. Values were calculated based on multiple sequence alignments across all samples.

Type	Length (bp)		S		Hap		Hd		π	
	Mean	SD	Mean	SD	Mean	SD	Mean	SD	Mean	SD
CDS	837.92	820.39	51.72	63.57	8.12	2.50	0.90	0.18	0.02	0.01
IGS	263.41	313.47	27.64	33.58	7.86	2.78	0.87	0.25	0.04	0.02
rRNA	1130.75	1197.19	8.00	11.29	4.50	3.42	0.54	0.42	0.00	0.00
tRNA	164.38	229.11	5.79	11.59	3.18	2.99	0.35	0.36	0.01	0.02
Total	459.41	633.80	31.50	45.90	7.17	3.24	0.79	0.32	0.03	0.02

## Data Availability

The plastomes sequence data that supports the findings of this study are openly available at figshare.com (https://doi.org/10.6084/m9.figshare.30038143, accessed on 25 August 2025).

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
