# Peer review of "Exploring Plastome Diversity and Molecular Evolution Within Genus Tortula (Family Pottiaceae, Bryophyta)"

_plants, 2025, doi:10.3390/plants14172808_

Round 1

Reviewer 1 Report

Comments and Suggestions for Authors

The article is a valuable contribution to the study of Pottiaceae genomics and is suitable for publication in Plants following some adjustments.

  1. The caption for Figure 5 (lines 346–354) requires correction. The purpose of the list of loci on the left is unclear. What does the phrase "Genes marked in red (cemA, psbB, rpoB) show variation associated with the phylogenetic placement of T. [Tortula] atrovirens" mean?
  2. The authors’ interpretation of their data requires clarification. The authors write (lines 461–464): "Our phylogenetic analysis confirmed the monophyly of nine Tortula species, with S. [Syntrichia] princeps as a well-supported outgroup. Nevertheless, it does not reflect the segregation of two of the sections proposed by Zander [5], namely, section Tortula and section Pottia. Only section Pottia appears to be monophyletic." However, Figure 5 shows two clades (monophyletic groups)—one corresponds to sect. Tortula, and the other to section Pottia. Furthermore, in which specific analysis did the authors test the monophyly of the nine species of section Tortula? None of the species in the study, as summarized in Figure 5, is represented by two or more samples. On what basis was the outgroup chosen (lines 127–128)? Instead of "... with S. princeps as a well-supported outgroup," it would be better to write "... with S. princeps as an outgroup."
  3. Lines 121–132. The authors’ taxonomic sampling should be discussed in much greater detail. Why exactly nine species? How many species in each section are recognized in the treatment by R.H. Zander? How many synonyms for each species are listed by this taxonomist? Etc. What is the meaning of the authors’ phylogenetic analysis based on nine species, if the number of species in the genus varies from 144 to 165 (lines 74–75)? In my opinion, the authors should either refer to their phylogenetic results as strictly preliminary or exclude taxonomic and morphological questions from the article altogether, especially since the study's conclusions do not mention the taxonomy of Tortula or the phylogenetic results at all.
  4. The description of the phylogenetic analysis in the "Materials and Methods" section should be more detailed. It is unclear why the sentence "Phylogenetic relationships were inferred using a maximum likelihood (ML) approach under the GTR+G substitution model, implemented in MEGA11" (lines 496–498) is placed in the "Library Construction and Sequencing" section (line 492).
  5. The discussion of previous phylogenetic results related to the article's main topic should be more extensive. From the current text, it is unclear whether any phylogenetic studies have been conducted on this moss group.

Author Response

The caption for Figure 5 (lines 346–354) requires correction. The purpose of the list of loci on the left is unclear. What does the phrase "Genes marked in red (cemA, psbB, rpoB) show variation associated with the phylogenetic placement of T. [Tortula] atrovirens" mean?

Response: Thank you for the comment, a clarification was added at lines 343-344.

The authors’ interpretation of their data requires clarification. The authors write (lines 461–464): "Our phylogenetic analysis confirmed the monophyly of nine Tortula species, with S. [Syntrichia] princeps as a well-supported outgroup. Nevertheless, it does not reflect the segregation of two of the sections proposed by Zander [5], namely, section Tortula and section Pottia. Only section Pottia appears to be monophyletic." However, Figure 5 shows two clades (monophyletic groups)—one corresponds to sect. Tortula, and the other to section Pottia. Furthermore, in which specific analysis did the authors test the monophyly of the nine species of section Tortula? None of the species in the study, as summarized in Figure 5, is represented by two or more samples. On what basis was the outgroup chosen (lines 127–128)? Instead of "... with S. princeps as a well-supported outgroup," it would be better to write "... with S. princeps as an outgroup."

Response: We appreciate this valuable feedback. We have revised the relevant paragraph to clarify that monophyly was assessed visually using the ML tree by identifying whether a clade could be separated with a single branch cut from the outgroup (S. princeps). Our intent was to observe phylogenetic patterns rather than make formal taxonomic conclusions. We have removed the term "well-supported" and now describe S. princeps simply as the outgroup, as suggested. While our tree shows distinct clades corresponding to sections Tortula and Pottia, only the latter is consistently monophyletic based on the sampled taxa. The purpose of the analysis was exploratory, focusing on phylogenetic signal from the chloroplast genome rather than resolving sectional classifications.

Lines 121–132. The authors’ taxonomic sampling should be discussed in much greater detail. Why exactly nine species? How many species in each section are recognized in the treatment by R.H. Zander? How many synonyms for each species are listed by this taxonomist? Etc. What is the meaning of the authors’ phylogenetic analysis based on nine species, if the number of species in the genus varies from 144 to 165 (lines 74–75)? In my opinion, the authors should either refer to their phylogenetic results as strictly preliminary or exclude taxonomic and morphological questions from the article altogether, especially since the study's conclusions do not mention the taxonomy of Tortula or the phylogenetic results at all.

Response: Thank you for this important point. We acknowledge that the genus Tortula comprises over 140 species. However, our study is focused on exploring chloroplast genomic diversity rather than conducting a taxonomic revision. The aim was to evaluate variability across genomes to identify loci that could serve as robust phylogenetic markers in future studies. Due to experimental constraints, including successful cultivation and DNA quality, only nine species yielded data of sufficient quality for chloroplast genome assembly (Clarification was added at M&M). While the sampling is limited, we consider the findings a valuable preliminary step toward understanding molecular evolution in the group. We now emphasize this in the manuscript and clarify that our conclusions are not taxonomic but methodological concluding the usefulness of each marker vs the whole chloroplast genome.

  1. The description of the phylogenetic analysis in the "Materials and Methods" section should be more detailed. It is unclear why the sentence "Phylogenetic relationships were inferred using a maximum likelihood (ML) approach under the GTR+G substitution model, implemented in MEGA11" (lines 496–498) is placed in the "Library Construction and Sequencing" section (line 492).

Response: Thank you for pointing this out. We have discarded the phylogenetic analysis sentence mistakenly written at lines 496-498.

The discussion of previous phylogenetic results related to the article's main topic should be more extensive. From the current text, it is unclear whether any phylogenetic studies have been conducted on this moss group.

Response: We appreciate this suggestion. In response, we have added a paragraph discussing the most relevant prior phylogenetic study on Tortula and related genera. We compare our findings with those of previous work, noting both congruent and divergent relationships. Additionally, we highlight the relevance of choosing chloroplast loci that offer higher resolution and suggest that discrepancies may reflect differences in marker selection or taxon sampling.

Reviewer 2 Report

Comments and Suggestions for Authors

This manuscript by Hassanezhad et al. presents a comparative analysis of chloroplast genomes across nine Tortula species. The study reveals conserved quadripartite plastome structures with size variations, identifies AT-rich SSRs and higher-order SSRs as potential population markers, and demonstrates non-monophyly in Tortula section Tortula through phyloplastomic analyses. Positive selection signals in specific genes are reported, providing a foundational plastomic resource for Pottiaceae and advancing our understanding of genomic adaptations in desiccation-tolerant mosses. While the work offers novel molecular markers for bryophyte phylogenetics and insights into environmental adaptation, the findings remain preliminary with incremental contributions to the field. Detailed comments and suggestions follow:

  1. It is recommended to provide additional background information on the phylogeny of Tortula in the Introduction section.
  2. The logical flow of the Introduction could be improved. For instance, consider moving the statement "Tortula is ecologically significant, as.." (Lines 115-120) to the fourth paragraph.
  3. While the authors sequenced the chloroplast genomes of nine Tortula species, the rationale for selecting these specific species is not clearly explained. Please clarify how these species contribute to resolving Tortula's phylogenetic relationships.
  4. It is recommended to include morphological images of the nine studied species. This would facilitate a better understanding of their phylogenetic relationships and morphological differences.
  5. The results of chloroplast genome assembly and annotation are not presented. Key aspects such as assembly completeness, annotation accuracy, and any unique genomic features observed should be included to demonstrate the study's significance and methodological rigor.
  6. The basis for selecting the three species labeled as "exemplary Tortula species" in Figure 1 is unclear. Please provide justification for this selection. Additionally, the chloroplast genome maps for the remaining species should be supplied in the Supplementary Materials.
  7. The results stating "This approach yielded a total of 90 candidate regions, comprising 49 IGS, 38 CDS, and three tRNA genes" (Lines 312-313) should be detailed in the Supplementary Materials.
  8. The description of the matrix used for phylogenetic analysis is ambiguous. Clarify whether the complete chloroplast genomic sequences (Line 322) or a concatenated matrix of specific regions (Line 351) was used.

Additionally, the description "atpE, atpI, cemA, chlN, psbB, matK, ndhA, ndhD, ndhD-trnL, rbcL, rpoB, rpoC1, rps7-trnV, and the structural regions LSC, SSC, and IRs" (Lines 325-326) is inaccurate. These genes/intergenic spacers are components of the LSC, SSC, or IR regions, not separate entities.

  1. In the phylogenetic analysis, only the Maximum Likelihood method was used for phylogenetic inference. To strengthen the analysis, the authors should consider adding Bayesian Inference as an additional method and comparing the results with those from ML.
  2. In Table 1, please merge Columns 1-2 into a single column titled "Species" with binomial nomenclature.
  3. In Figure 6, the title of the X-axis should be modified, and "omega" should be replaced with the standard symbol "ω".
  4. The primary references for the analysis software mentioned in the Materials and Methods should be provided. e.g. FastQC (Line500), CPGAVAS2 (Line503), IRscope (Line519).
  5. The description of the phylogenetic tree construction methodology is inconsistent: “Phylogenetic relationships were inferred using a maximum likelihood (ML) approach under the GTR+G substitution model, implemented in MEGA11.” (Line 496-498), and “Phylogenetic reconstruction was initially performed using FastTree v2 with the Generalized Time Reversible (GTR)” (Line 555-556).

Author Response

This manuscript by Hassanezhad et al. presents a comparative analysis of chloroplast genomes across nine Tortula species. The study reveals conserved quadripartite plastome structures with size variations, identifies AT-rich SSRs and higher-order SSRs as potential population markers, and demonstrates non-monophyly in Tortula section Tortula through phyloplastomic analyses. Positive selection signals in specific genes are reported, providing a foundational plastomic resource for Pottiaceae and advancing our understanding of genomic adaptations in desiccation-tolerant mosses. While the work offers novel molecular markers for bryophyte phylogenetics and insights into environmental adaptation, the findings remain preliminary with incremental contributions to the field. Detailed comments and suggestions follow:

It is recommended to provide additional background information on the phylogeny of Tortula in the Introduction section.

Response: The introduction was modified as requested.

The logical flow of the Introduction could be improved. For instance, consider moving the statement "Tortula is ecologically significant, as.." (Lines 115-120) to the fourth paragraph.

Response: The statement was positioned as requested.

While the authors sequenced the chloroplast genomes of nine Tortula species, the rationale for selecting these specific species is not clearly explained. Please clarify how these species contribute to resolving Tortula's phylogenetic relationships.

Response: Thank you for your comment. However, we would like to clarify that this study is not intended as a taxonomic revision to resolve Tortula phylogenetic relationships but rather aims to provide a foundation for selecting suitable chloroplast markers that can capture the phylogenetic signal of the full plastome. The rationale for using these specific species is explained in the Introduction (last paragraph), the Discussion (last paragraph, where we acknowledge the limited number of species and its implications for taxonomic resolution), and the Materials and Methods section (sampling), where we detail the criteria used for species selection.

It is recommended to include morphological images of the nine studied species. This would facilitate a better understanding of their phylogenetic relationships and morphological differences.

Response: Thank you for the suggestion. As this study is primarily focused on chloroplast genome variation and the identification of phylogenetically informative loci, we did not emphasize morphological comparisons or taxonomic revision. For this reason, morphological imaging was not included. However, we agree that such images may be valuable in future taxonomic or integrative studies combining genomic and morphological data.

The results of chloroplast genome assembly and annotation are not presented. Key aspects such as assembly completeness, annotation accuracy, and any unique genomic features observed should be included to demonstrate the study's significance and methodological rigor.

Response: This part was clarified and detailed in results part 2.1.

The basis for selecting the three species labeled as "exemplary Tortula species" in Figure 1 is unclear. Please provide justification for this selection. Additionally, the chloroplast genome maps for the remaining species should be supplied in the Supplementary Materials.

Response: As gene content and synteny are the same, we didn’t include additional maps, we selected three from different clades to demonstrate that fact, and due to figure resolution, we didn’t include more than three, thus the figure will be readable.

The results stating "This approach yielded a total of 90 candidate regions, comprising 49 IGS, 38 CDS, and three tRNA genes" (Lines 312-313) should be detailed in the Supplementary Materials.

Response: Table S3 is already cited at the beginning of this section, however as instructed we recited the supplementary table at the end of this statement.

The description of the matrix used for phylogenetic analysis is ambiguous. Clarify whether the complete chloroplast genomic sequences (Line 322) or a concatenated matrix of specific regions (Line 351) was used.

Additionally, the description "atpE, atpI, cemA, chlN, psbB, matK, ndhA, ndhD, ndhD-trnL, rbcL, rpoB, rpoC1, rps7-trnV, and the structural regions LSC, SSC, and IRs" (Lines 325-326) is inaccurate. These genes/intergenic spacers are components of the LSC, SSC, or IR regions, not separate entities.

Response: The phylogenetic analysis was performed using the complete chloroplast genome (not concatenated regions), as well as individual loci and the entire LSC, SSC, and IR regions separately. Therefore, the use of specific markers located within the LSC, for example, differs from analyzing the entire LSC region as a whole. Conducting phylogenetic analyses based on the LSC, SSC, and IR regions individually allows for the assessment of both evolutionary homogeneity and divergence among the major structural components of the chloroplast genome. We rephrased the sentence to improve accuracy in the revised manuscript.

In the phylogenetic analysis, only the Maximum Likelihood method was used for phylogenetic inference. To strengthen the analysis, the authors should consider adding Bayesian Inference as an additional method and comparing the results with those from ML.

Response: Thank you for the suggestion. We conducted a Bayesian inference (BI) analysis of the full plastome dataset, which produced a tree topology identical to that of the Maximum Likelihood (ML) analysis, with similarly strong support across nodes. Given this congruence and the exploratory nature of our study—which focuses on identifying loci that reproduce the full plastome phylogenetic signal rather than on taxonomic resolution—we chose to use ML consistently for the individual loci. We believe this approach is appropriate for demonstrating marker compatibility and phylogenetic consistency. However, we remain open to including BI results if the reviewer deems it critical use.

In Table 1, please merge Columns 1-2 into a single column titled "Species" with binomial nomenclature.

Response: Modified as instructed.

In Figure 6, the title of the X-axis should be modified, and "omega" should be replaced with the standard symbol "ω".

Response: Modified as instructed.

The primary references for the analysis software mentioned in the Materials and Methods should be provided. e.g. FastQC (Line500), CPGAVAS2 (Line503), IRscope (Line519).

Response: References was provided as a link for the software mainpage for FastQC, tool name was corrected to GeSeq and cited for CPGAVAS2, and the reference was provided for IRscope.

The description of the phylogenetic tree construction methodology is inconsistent: “Phylogenetic relationships were inferred using a maximum likelihood (ML) approach under the GTR+G substitution model, implemented in MEGA11.” (Line 496-498), and “Phylogenetic reconstruction was initially performed using FastTree v2 with the Generalized Time Reversible (GTR)” (Line 555-556).

Response: This is a mistake, and it was corrected in the revised version, discarding the statement of MEGA11 and its reference.

Reviewer 3 Report

Comments and Suggestions for Authors

ABSTRACT

  • It is implied that the study aimed to analyze plastomes, but it is not clearly stated what scientific questions were asked or what hypotheses were tested.
  • The relevance of the findings and how they contribute to the understanding of evolution in Tortula or Pottiaceae is not clearly summarized.
  • Only essential methods should be retained in the abstract: NGS, plastome analysis, codon usage, phylogeny — without platform details or statistical values.
  • Rearrange the information in a logical order: introduction (context & problem); objective(s); brief methodology; key results (without excessive numbers); conclusions / relevance.
  • Add a clear and academic closing sentence.

INTRODUCTION

  • Although useful for taxonomic context, the introduction spends too much space describing moss morphology and sporophyte characters (e.g., peristome, midrib), which would be more appropriate in the "Materials and Methods" section or a comparative taxonomy subchapter. I suggest compressing taxonomic details.
  • The text provides extensive general context on bryophytes but does not clearly specify what knowledge gaps exist and why studying Tortula plastomes is relevant. This gap should be explicitly stated. Mention early (in paragraph 1 or 2) why plastomes are a relevant tool for phylogeny and adaptation.
  • Information about chloroplasts and the importance of plastomes in systematics appears only after a full page of morphological and classification details. A reordering is needed to reflect the study’s true focus: genomics, not classical taxonomy.
  • Although there is a sentence indicating that “plastome diversity is underexplored,” no direct scientific question or explicit hypothesis is formulated.

MATERIALS AND METHODS

  • The methodology is technically sound but presented as a compact block. It lacks clear subtitles or a visible logical structure (e.g., “Sampling”, “Sequencing and Assembly”, “Bioinformatic Analyses”).
  • It is stated that only one accession per species was ultimately analyzed, but it is not scientifically justified whether this is representative. This may raise concerns about the robustness of the results. You can mention logistical constraints or that intraspecific variation was beyond the scope of the study.
  • The software used (Geneious, SPAdes, MEGA, etc.) is listed, but key parameters are not specified (e.g., k-mers for assembly, alignment methods, thresholds for SSR selection).
  • There is no mention of any assembly validation (e.g., comparison with reference plastomes, read realignment), which is essential for methodological rigor.

RESULTS

  • The results are overloaded with exact values (e.g., lengths in bp, RSCU values with many decimals, SSR motif frequencies), which overwhelms the reader and obscures key ideas. For example, reporting every 1–2 bp variation in IR or LSC regions weakens the impact of the conclusions. Focus on trends and synthesized ideas rather than every figure. For instance: “Plastome sizes vary slightly among species, with an overall conserved structure and minor differences in IR and LSC regions.”
  • Molecular markers are not clearly highlighted as a central finding. While some informative genes (e.g., matK, ATPE, ndhD) are mentioned, their comparative value or recommended use (e.g., barcoding, phylogeny, selection studies) is not made explicit. Organize the identified markers more clearly!
  • The description of tables is nearly copied into the text without synthesis. For instance, every GC content and genome length is restated despite being in a table.

DISCUSSION

  • Although numerous references and comparisons with literature are included, some paragraphs merely restate the results or juxtapose other studies without drawing clear conclusions or formulating interpretive hypotheses. The discussion is well documented, but sometimes descriptive rather than analytical. Actively compare studies — don't just list them.
  • Many aspects are discussed (plastome size, codon usage, SSRs), but the initial research question is not explicitly revisited, and it is unclear to what extent it was convincingly addressed.
  • The observation regarding the polyphyly of the Tortula section is mentioned, but not explored in depth: what does this mean for taxonomy? What classification changes might be justified?
  • The opportunity to highlight the study’s original contributions is missed. Some findings (e.g., matK and ATPE markers supported by positive selection) are treated as simple observations, though they are highly relevant for phylogeny and adaptation.
  • Add a paragraph addressing limitations and future directions.

CONCLUSIONS

  • The conclusions reiterate too many results without synthesis. Rather than summarizing the core ideas and the study’s impact, the section reads more like a fragmented recap of previous sections.
  • There are no clear statements regarding the original contributions.
  • Markers and plastome structure are mentioned, but it is not explained how they could be used in practice (e.g., in systematics, molecular identification, conservation, or adaptation studies).
  • There is no final sentence that clearly expresses the importance or overall message of the study.

Author Response

ABSTRACT

  • It is implied that the study aimed to analyze plastomes, but it is not clearly stated what scientific questions were asked or what hypotheses were tested.
  • The relevance of the findings and how they contribute to the understanding of evolution in Tortula or Pottiaceae is not clearly summarized.
  • Only essential methods should be retained in the abstract: NGS, plastome analysis, codon usage, phylogeny — without platform details or statistical values.
  • Rearrange the information in a logical order: introduction (context & problem); objective(s); brief methodology; key results (without excessive numbers); conclusions / relevance.
  • Add a clear and academic closing sentence.

Response: The abstract was rewritten to include all the reviewer comments.

INTRODUCTION

  • Although useful for taxonomic context, the introduction spends too much space describing moss morphology and sporophyte characters (e.g., peristome, midrib), which would be more appropriate in the "Materials and Methods" section or a comparative taxonomy subchapter. I suggest compressing taxonomic details.
  • The text provides extensive general context on bryophytes but does not clearly specify what knowledge gaps exist and why studying Tortula plastomes is relevant. This gap should be explicitly stated. Mention early (in paragraph 1 or 2) why plastomes are a relevant tool for phylogeny and adaptation.
  • Information about chloroplasts and the importance of plastomes in systematics appears only after a full page of morphological and classification details. A reordering is needed to reflect the study’s true focus: genomics, not classical taxonomy.
  • Although there is a sentence indicating that “plastome diversity is underexplored,” no direct scientific question or explicit hypothesis is formulated.

Response: Thank you for the detailed suggestions. We have revised the Introduction to reduce the morphological detail and initially relocated some taxonomic content to the Materials and Methods section, but finally was discarded it form the text. While the overall flow—beginning with organismal context, transitioning to molecular studies, and culminating in plastome-based approaches—has been preserved, we have made the text more concise and clearly aligned with the genomic focus of the study. The challenges of plastome sequencing and the relevance of plastome data is now introduced clearer, and we have explicitly stated the study’s aim in the final paragraph: to assess plastome diversity in Tortula and to identify variable loci that may serve as cost-effective phylogenetic markers. These changes address the identified gaps and clarify the rationale and objectives guiding our work.

MATERIALS AND METHODS

  • The methodology is technically sound but presented as a compact block. It lacks clear subtitles or a visible logical structure (e.g., “Sampling”, “Sequencing and Assembly”, “Bioinformatic Analyses”).
  • It is stated that only one accession per species was ultimately analyzed, but it is not scientifically justified whether this is representative. This may raise concerns about the robustness of the results. You can mention logistical constraints or that intraspecific variation was beyond the scope of the study.
  • The software used (Geneious, SPAdes, MEGA, etc.) is listed, but key parameters are not specified (e.g., k-mers for assembly, alignment methods, thresholds for SSR selection).
  • There is no mention of any assembly validation (e.g., comparison with reference plastomes, read realignment), which is essential for methodological rigor.

Response: We thank the reviewer for these thoughtful suggestions. In the revised manuscript, the "Materials and Methods" section has been restructured with clear subtitles to improve readability and logical flow. We clarified that only one accession per species was used due to the technical limitations in obtaining sufficient high-quality DNA from moss samples for NGS, and that intraspecific variation was beyond the current study’s scope. Methodological details have been expanded to include key parameters such as SPAdes k-mer values, SSR detection thresholds, and the number of read-mapping iterations for consensus refinement. We also added that plastome assemblies were validated by reassembling with SPAdes using trusted contigs and inspecting circularity and completeness within Geneious, and the annotation was further based on Syntrichia ruralis in chlorobox online tool. These revisions strengthen the methodological rigor and transparency of our work.

RESULTS

  • The results are overloaded with exact values (e.g., lengths in bp, RSCU values with many decimals, SSR motif frequencies), which overwhelms the reader and obscures key ideas. For example, reporting every 1–2 bp variation in IR or LSC regions weakens the impact of the conclusions. Focus on trends and synthesized ideas rather than every figure. For instance: “Plastome sizes vary slightly among species, with an overall conserved structure and minor differences in IR and LSC regions.”
  • Molecular markers are not clearly highlighted as a central finding. While some informative genes (e.g., matK, ATPE, ndhD) are mentioned, their comparative value or recommended use (e.g., barcoding, phylogeny, selection studies) is not made explicit. Organize the identified markers more clearly!
  • The description of tables is nearly copied into the text without synthesis. For instance, every GC content and genome length is restated despite being in a table.

Response: We thank the reviewer for the comment. While we agree that excessive numeric detail can obscure broader trends, we aimed to strike a balance between conciseness and the methodological expectations of plastome-based comparative studies. Key values (e.g., plastome lengths, GC content, IR boundaries) were included selectively to support meaningful interspecific comparisons and highlight subtle but evolutionarily relevant structural differences. These elements are commonly used to infer evolutionary patterns in moss plastomes and are particularly important in non-model taxa like Tortula. We respectfully believe that the manuscript does not present an excessive number of exact values. Numerical data were included selectively and primarily in summary form (e.g., means, ranges, standard deviations), specifically to support meaningful biological observations such as plastome length differences, and GC%. Details like 1–2 bp variations in IR or LSC regions were only mentioned when evolutionarily relevant. Regarding RSCU values, only codons with notable usage bias across 11 amino acids and stop codons were highlighted, not an exhaustive complete codon list. Similarly, SSR results were synthesized to show trends in motif abundance and taxon-specific variation, not every motif frequency.

As for molecular markers, their identification was clearly outlined in Sections 2.5.2 and 2.5.3, where they were selected based on haplotype diversity, phylogenetic signal, and selection pressure (dN/dS ratio). These markers were also summarized in Figure 5 and referenced explicitly in the Abstract and Discussion, particularly atpE and matK as high-priority candidates. Lastly, we did not restate all values from tables in the text and included only the most relevant numerical insights. We therefore believe that the results are presented in a focused and biologically meaningful manner, however we are gladly open for specific suggestions.

DISCUSSION

  • Although numerous references and comparisons with literature are included, some paragraphs merely restate the results or juxtapose other studies without drawing clear conclusions or formulating interpretive hypotheses. The discussion is well documented, but sometimes descriptive rather than analytical. Actively compare studies — don't just list them.
  • Many aspects are discussed (plastome size, codon usage, SSRs), but the initial research question is not explicitly revisited, and it is unclear to what extent it was convincingly addressed.
  • The observation regarding the polyphyly of the Tortula section is mentioned, but not explored in depth: what does this mean for taxonomy? What classification changes might be justified?
  • The opportunity to highlight the study’s original contributions is missed. Some findings (e.g., matK and ATPE markers supported by positive selection) are treated as simple observations, though they are highly relevant for phylogeny and adaptation.
  • Add a paragraph addressing limitations and future directions.

Response: We appreciate the insightful feedback. The discussion has been revised to emphasize analytical interpretations over descriptive restatement. We clarified the study’s original aim and its fulfillment by highlighting the identification of lineage-specific markers (e.g., atpE, matK) and their relevance to phylogeny and adaptive evolution. The implications of polyphyly within Tortula were elaborated, suggesting a re-evaluation of traditional sectional classification based on plastome data. We also added a concluding paragraph outlining limitations and highlighted particularly the use of single accessions when interpreting the phylogenetic signal. The future directions involving broader taxon sampling and intraspecific validation were also added.

CONCLUSIONS

  • The conclusions reiterate too many results without synthesis. Rather than summarizing the core ideas and the study’s impact, the section reads more like a fragmented recap of previous sections.
  • There are no clear statements regarding the original contributions.
  • Markers and plastome structure are mentioned, but it is not explained how they could be used in practice (e.g., in systematics, molecular identification, conservation, or adaptation studies).
  • There is no final sentence that clearly expresses the importance or overall message of the study.

Response: Thank you for the constructive feedback. We have revised the Conclusions section to better synthesize the results and highlight the study’s original contributions. Specifically, we emphasized how key plastome regions such as matK and atpE can be applied in phylogenetics and adaptation studies, and we outlined additional markers that expand the molecular toolkit for systematics and conservation. We also included a conclusive statement underscoring the broader relevance and future application of our findings in bryophyte research. The revised section now presents a more cohesive and impactful summary of the study’s contributions and practical implications.

Round 2

Reviewer 1 Report

Comments and Suggestions for Authors

 The article is acceptable in the present form.

Author Response

The authors thank the anonymous reviewer 1 for his comments, which have significantly improved the content of the manuscript.

Reviewer 2 Report

Comments and Suggestions for Authors

Comments to the Author

The authors have made further improvements in this revision, and several reviewer concerns appear adequately addressed. However, the following specific issues require attention to enhance clarity and scientific rigor:

1. Regarding my previous comment “It is recommended to include morphological images of the nine studied species. This would facilitate a better understanding of their phylogenetic relationships and morphological differences.”

      I fully understand the authors' response stating "this study is not intended as a taxonomic revision to resolve Tortula phylogenetic relationships." However, as indicated by the title focusing partly on "Molecular Evolution," morphological images of the nine studied species would significantly aid readers in understanding the evolutionary significance of these taxa and broaden the paper's appeal. Moreover, accurate phylogenetic reconstruction forms the essential foundation for understanding plastome evolution. If the scope is indeed limited solely to "the identification of phylogenetically informative loci" as stated in the response, I recommend revising the title of the manuscript to reflect this narrower focus accurately.

2. Regarding my previous comment “The results of chloroplast genome assembly and annotation are not presented. Key aspects such as assembly completeness, annotation accuracy, and any unique genomic features observed should be included to demonstrate the study's significance and methodological rigor.”

      While the authors have presented some assembly/annotation results, detailed information on quality control procedures for the raw data is still required. This must include specifics on pre-assembly filtering steps and/or post-assembly inspection and correction methods applied to the results. This information remains crucial for assessing the reliability and accuracy of the genomic data.

3. Regarding my previous comment “The basis for selecting the three species labeled as "exemplary Tortula species" in Figure 1 is unclear. Please provide justification for this selection. Additionally, the chloroplast genome maps for the remaining species should be supplied in the Supplementary Materials.”

      I maintain my strong recommendation that the chloroplast genome maps for the remaining six species be provided in the Supplementary Materials, unless their genomic structures are demonstrably identical to the three "exemplary" species shown in Figure 1. Providing all maps enhances transparency and allows readers to assess structural variation fully.

4. Regarding my previous comment “The description of the matrix used for phylogenetic analysis is ambiguous. Clarify whether the complete chloroplast genomic sequences (Line 322) or a concatenated matrix of specific regions (Line 351) was used.”

      Based on the authors' response clarifying the matrix usage, I request that the phylogenetic trees resulting from the different matrices (complete chloroplast genomes vs. concatenated specific regions) be included in the Supplementary Materials. This allows readers to directly compare the results obtained from these distinct datasets.

5. Regarding my previous comment “In the phylogenetic analysis, only the Maximum Likelihood method was used for phylogenetic inference. To strengthen the analysis, the authors should consider adding Bayesian Inference as an additional method and comparing the results with those from ML.”

      Given the authors' confirmation that the BI tree topology is identical to the ML tree, the posterior probabilities from the BI analysis should be added alongside the ML bootstrap values on the nodes of Figure 5. And the full BI tree should be presented in the Supplementary Materials. Additionally, please ensure the bootstrap values on the ML tree in Figure 5 are presented as percentages.

Author Response

The authors thank the anonymous reviewer 2 for his comments, which have significantly improved the content of the manuscript.

Comment 1

Regarding my previous comment “It is recommended to include morphological images of the nine studied species. This would facilitate a better understanding of their phylogenetic relationships and morphological differences.”

      I fully understand the authors' response stating "this study is not intended as a taxonomic revision to resolve Tortula phylogenetic relationships." However, as indicated by the title focusing partly on "Molecular Evolution," morphological images of the nine studied species would significantly aid readers in understanding the evolutionary significance of these taxa and broaden the paper's appeal. Moreover, accurate phylogenetic reconstruction forms the essential foundation for understanding plastome evolution. If the scope is indeed limited solely to "the identification of phylogenetically informative loci" as stated in the response, I recommend revising the title of the manuscript to reflect this narrower focus accurately.

Response:
We appreciate the reviewer’s thoughtful suggestion. As previously mentioned, the focus of this study is not taxonomic revision, nor does it aim to reconstruct phylogenetic relationships based on morphological traits. Rather, the primary objective is a comparative chloroplast genomics analysis of Tortula, emphasizing genome structure and the identification of phylogenetically informative plastid loci.

While the term “molecular evolution” appears in the title, it specifically refers to sequence-level divergence, codon usage patterns, SSR distribution, and selection signals across plastomes—not to evolutionary inferences based on morphology. For this reason, we respectfully believe that the current title accurately reflects the study’s aims and does not require adjustment.

Furthermore, own morphological images were not included because the specimens were selected for their capsules and gametophytes were cultivated axenically for DNA extraction. As such, we do not currently possess high-resolution, publication-quality images of all nine species suitable for inclusion. Including them now would divert from the genomic focus of the manuscript. Nevertheless, with the aim of facilitating species identification for readers, as proposed by the reviewer, we provide literature references and web pages where detailed descriptions, illustrations, and photographs can be found.

Comment 2

Regarding my previous comment “The results of chloroplast genome assembly and annotation are not presented. Key aspects such as assembly completeness, annotation accuracy, and any unique genomic features observed should be included to demonstrate the study's significance and methodological rigor.”

      While the authors have presented some assembly/annotation results, detailed information on quality control procedures for the raw data is still required. This must include specifics on pre-assembly filtering steps and/or post-assembly inspection and correction methods applied to the results. This information remains crucial for assessing the reliability and accuracy of the genomic data.

Response:
Thank you for this important note. The “Plastome Assembly and Annotation section” has been revised to include specific details on quality control and data validation. We now explicitly describe pre-assembly steps (FastQC, Trimmomatic), the assembly pipeline (Geneious with SPAdes confirmation using multiple k-mers), post-assembly steps (five rounds of read re-mapping, consensus extraction), IR detection, and manual curation. These updates demonstrate our commitment to methodological rigor and ensure the robustness of our genomic data.

Comment 3

Regarding my previous comment “The basis for selecting the three species labeled as "exemplary Tortula species" in Figure 1 is unclear. Please provide justification for this selection. Additionally, the chloroplast genome maps for the remaining species should be supplied in the Supplementary Materials.”

      I maintain my strong recommendation that the chloroplast genome maps for the remaining six species be provided in the Supplementary Materials, unless their genomic structures are demonstrably identical to the three "exemplary" species shown in Figure 1. Providing all maps enhances transparency and allows readers to assess structural variation fully.

Response:
We have now provided Figures S1 in the Supplementary Materials, which contains chloroplast genome maps for all nine Tortula species, not just the three previously shown.

Comment 4

Regarding my previous comment “The description of the matrix used for phylogenetic analysis is ambiguous. Clarify whether the complete chloroplast genomic sequences (Line 322) or a concatenated matrix of specific regions (Line 351) was used.”

      Based on the authors' response clarifying the matrix usage, I request that the phylogenetic trees resulting from the different matrices (complete chloroplast genomes vs. concatenated specific regions) be included in the Supplementary Materials. This allows readers to directly compare the results obtained from these distinct datasets.

Response:
We clarified the methodology to indicate that both full plastome alignments and concatenated matrices of selected loci were used in phylogenetic reconstruction. To allow direct comparison of these analyses, we included the resulting phylogenetic trees from all datasets (complete plastomes, LSC, SSC, IR, and concatenated regions) in Supplementary File S2.

Comment 5
Regarding my previous comment “In the phylogenetic analysis, only the Maximum Likelihood method was used for phylogenetic inference. To strengthen the analysis, the authors should consider adding Bayesian Inference as an additional method and comparing the results with those from ML.”

      Given the authors' confirmation that the BI tree topology is identical to the ML tree, the posterior probabilities from the BI analysis should be added alongside the ML bootstrap values on the nodes of Figure 5. And the full BI tree should be presented in the Supplementary Materials. Additionally, please ensure the bootstrap values on the ML tree in Figure 5 are presented as percentages.

Response:

We thank the reviewer for encouraging these improvements, which have enhanced both the analytical rigor and clarity of our phylogenetic interpretations. During this revision, we re-evaluated the placement of T. atrovirens across individual loci and identified a mislabeling of the deviating markers. This has now been corrected throughout the manuscript. During the revision and the re-evaluation we have reconducted the Bayesian Inference (BI) analyses using extended MCMC runs of 2,000,000 generations. The resulting topologies are congruent with those obtained from the Maximum Likelihood (ML) approach. In response to the reviewer’s suggestion, we have added BI posterior probabilities alongside the ML bootstrap values on the nodes of Figure 5. Please note that the ML bootstrap values are presented in decimal format (0–1) due to the output convention of FastTree, which reports support values as proportions rather than percentages.

Additionally, we have included the full BI trees for all datasets—complete plastomes, LSC, SSC, IR regions, the concatenated hypervariable loci, and the two individual loci (atpI and ndhD) that exhibited variation—in Supplementary File S2 to support comparison and transparency.

Reviewer 3 Report

Comments and Suggestions for Authors

In my opinion, the paper can be published in its current form.

Author Response

The authors thank the anonymous reviewer 3 for his comments, which have significantly improved the content of the manuscript.

Round 3

Reviewer 2 Report

Comments and Suggestions for Authors

In this revised version, the authors further improved the analyses. I think most of the concerns raised by reviews have been adequately addressed and the manuscript has been improved.

During the phylogenetic analysis, only two concatenated matrices of protein-coding genes were analyzed (Figs. S2E and S2F). Therefore, the manuscript's claim of analyzing 'individual loci' (Lines 326-327 & 338-340) is inaccurate. The authors should either revise the text to remove references to individual locus analysis or provide supplementary phylogenetic trees for each individual locus.

Author Response

Comment

In this revised version, the authors further improved the analyses. I think most of the concerns raised by reviews have been adequately addressed and the manuscript has been improved.

During the phylogenetic analysis, only two concatenated matrices of protein-coding genes were analyzed (Figs. S2E and S2F). Therefore, the manuscript's claim of analyzing 'individual loci' (Lines 326-327 & 338-340) is inaccurate. The authors should either revise the text to remove references to individual locus analysis or provide supplementary phylogenetic trees for each individual locus.

Response:

Thank you very much for this insightful comment. We carefully revised all phylogenetic trees and ensured a more precise description of the analyses. In addition, we updated Supplementary Figure S2 to include more detailed information. Specifically, Fig. S2E is now expanded as Fig. S2.2, and Fig. S2F as Fig. S2.3, both expanded to include the individual trees for each of the selected loci. The necessary clarifications were also incorporated into the Results section and the corresponding figure legends in the revised manuscript.